# Peptides Derived from (RRWQWRMKKLG)_2-_K-*Ahx* Induce Selective Cellular Death in Breast Cancer Cell Lines through Apoptotic Pathway

**DOI:** 10.3390/ijms21124550

**Published:** 2020-06-26

**Authors:** Diego Sebastián Insuasty-Cepeda, Andrea Carolina Barragán-Cárdenas, Alejandra Ochoa-Zarzosa, Joel E. López-Meza, Ricardo Fierro-Medina, Javier Eduardo García-Castañeda, Zuly Jenny Rivera-Monroy

**Affiliations:** 1Departamento de Química, Facultad de Ciencias, Universidad Nacional de Colombia, Bogotá, Carrera 45 No 26-85, Building 451, office 409, Bogotá 11321, Colombia; dsinsuastyc@unal.edu.co (D.S.I.-C.); abarraganc@unal.edu.co (A.C.B.-C.); rfierrom@unal.edu.co (R.F.-M.); 2Facultad de Medicina Veterinaria y Zootecnia, Centro Multidisciplinario de Estudios en Biotecnología, Universidad Michoacana de San Nicolás de Hidalgo, Km 9.5 Carretera Morelia-Zinapécuaro, Posta Veterinaria, P.C. 58893 Morelia, Michoacán, Mexico; ochoaz@umich.mx (A.O.-Z.); elmeza@umich.mx (J.E.L.-M.); 3Departamento de Farmacia, Facultad de Ciencias, Universidad Nacional de Colombia, Bogotá Carrera 45 No 26-85, Building 450, Bogotá 11321, Colombia; jaegarciaca@unal.edu.co

**Keywords:** breast cancer, bovine lactoferricin, anticancer peptides, dimeric peptides

## Abstract

The effect on the cytotoxicity against breast cancer cell lines of the substitution of ^26^Met residue in the sequence of the Bovine Lactoferricin-derived dimeric peptide LfcinB (20-30)_2_: (^20^RRWQWRMKKLG^30^)_2_-K-*Ahx* with amino acids of different polarity was evaluated. The process of the synthesis of the LfcinB (20-30)_2_ analog peptides was similar to the original peptide. The cytotoxic assays showed that some analog peptides exhibited a significant cytotoxic effect against breast cancer cell lines HTB-132 and MCF-7, suggesting that the substitution of the Met with amino acids of a hydrophobic nature drastically enhances its cytotoxicity against HTB-132 and MCF-7 cells, reaching IC_50_ values up to 6 µM. In addition, these peptides have a selective effect, since they exhibit a lower cytotoxic effect on the non-tumorigenic cell line MCF-12. Interestingly, the cytotoxic effect is fast (90 min) and is maintained for up to 48 h. Additionally, through flow cytometry, it was found that the obtained dimeric peptides generate cell death through the apoptosis pathway and do not compromise the integrity of the cytoplasmic membrane, and there are intrinsic apoptotic events involved. These results show that the obtained peptides are extremely promising molecules for the future development of drugs for use against breast cancer.

## 1. Introduction

Cancer is considered to be the biggest public health problem worldwide [1], being the second most common cause of death in the world. In 2018, 8.2 million people died from this disease, and 14 million new cases are reported annually [2], with more than 60% occurring in Asia, Africa, and South America. In 2012, the most frequently diagnosed cancers in women were breast, colon, rectum, lung, cervix, and stomach, while for men they were lung, prostate, colon, rectum, stomach, and liver [3].

Breast cancer is the most common cancer type diagnosed worldwide [4]. There are 2.1 million cases annually, and 627,000 deaths of women due to this disease were reported in 2018 [5]. There are therapeutic options such as chemotherapy [6], radiotherapy [7], hormone therapy [8], and surgery [9] that have managed to mitigate this disease; however, these are invasive procedures with serious adverse effects that significantly affect the quality of life of the patients [10]. It is imperative to find new, more selective and less invasive therapeutic agents. In this context, several peptides derived from Bovine Lactoferricin (LfcinB) have been shown to have a selective cytotoxic effect against various cancer cell lines [11,12,13,14]. Some studies have shown that LfcinB-derived peptides such as dimers and tetramers containing the minimal motif RRWQWR exhibit a cytotoxic effect on oral squamous and breast cancer cell lines with IC_50_ values between 5 and 15 μM and do not exhibit a cytotoxic effect against non-tumorigenic cell lines such as PCS-201-012 and Het-1A [15,16]. The tetrameric peptide LfcinB (20-25)_4_: ((^20^RRWQWR^25^)_2_-K-*Ahx*-C)_2_ has exhibited a selective cytotoxic effect against MDA-MB 468, MDA-MB 231, and MCF-7 breast cancer cells and also induces cell death in MCF-7 cells through the apoptosis pathway [16,17]. Additionally, the dimeric peptide LfcinB (20-30)_2_: (^20^RRWQWRMKKLG^30^)_2_-K-*Ahx* has exhibited a significant selective cytotoxic effect against MDA-MB 468 and MDA-MB 231 breast cancer cells. The cytotoxic effect against both cell lines was near 100% (cellular viability ~0%) when the peptide concentration was 100 µg/mL (30 µM), suggesting that this peptide could be considered to be promising for the development of therapeutic agents for treating breast cancer.

In the present investigation, we wanted to identify analog peptides derived from the dimeric peptide LfcinB (20-30)_2_: (^20^RRWQWRMKKLG^30^)_2_-K-*Ahx* that have a greater selective cytotoxic effect against breast cancer cell lines than the original peptide. Additionally, we also sought to explore the possibility of changing the Met residue since it is prone to oxidation, and previously we had evidenced stability problems of LfcinB (20-30)_2_ during storage (Appendix A). For this, dimeric analog peptides containing the sequence LfcinB (20-30): (^20^RRWQWRMKKLG^30^) in which the amino acid ^26^Met (in bold and underlined) was replaced by amino acids such as Lys, Asp, Ala, Phe or Leu were synthesized, purified, and characterized.

The cytotoxic effect of analog peptides against breast cancer cell lines HTB-132 and MCF-7 and the non-tumorigenic cell line MCF-12 was evaluated. It was possible to identify analog peptides with higher selective cytotoxic effect against the breast cancer cell lines evaluated, these being peptides considered to be promising molecules for developing drugs for use against this type of cancer.

## 2. Results

In order to identify peptides derived from the dimeric peptide LfcinB (20-30)_2_ (referred to here as ^26^[M]) with enhanced anticancer activity against breast cancer cell lines, six dimeric peptides were synthesized via SPPS-Fmoc/tBu (as described in Appendix A), purified by means of RP-SPE, and characterized through RP-HPLC and MALDI-TOF MS (Appendix A), Table 1.

From the starting peptide ^26^[M], five new analog peptides were synthesized, in which the ^26^Met amino acid of the sequence was replaced by amino acids of a different nature: (i) Lys (basic residue), (ii) Asp (acid side chain), (iii) Ala and Leu (aliphatic side chains), and (iv) Phe (aromatic side chain). All the peptides were synthesized following the manual synthesis protocol established in our laboratory. The synthesis of analog peptides was similar to that of the original dimeric peptide ^26^[M], indicating that the substitution of amino acids does not affect the synthesis of the peptides. The peptides were obtained with high purity and had the expected mass, which was determined via MALDI-TOF MS (Table 1).

The cytotoxic effect of the peptide ^26^[M] against breast cancer cell lines HTB-132 and MCF-7 was evaluated (Figure 1). The HTB-132 cell line corresponds to triple-negative breast cancer that is one of the most aggressive types [18], and the MCF-7 cell line is a hormone-responsive cancer from a luminal type A breast cancer, which is the most diagnosed worldwide [19]. As is seen in Figure 1A, the peptide ^26^[M] exhibited a concentration-dependent cytotoxic effect against the HTB-132 cell line in the 1–100 µg/mL range of concentration, reaching a cytotoxic effect close to 55% at the maximum evaluated concentration. While the peptide ^26^[M] does not exhibit a significant cytotoxic effect against the MCF-7 cell line (Figure 1D), this behavior is consistent with other studies, where the MCF-7 line has been found to be less susceptible to treatments with LfcinB-derived peptides and to drugs such as doxorubicin [20].

The cytotoxic effect of analog dimeric peptides against the HTB-132 cell line was evaluated (Figure 1A–C). In a manner similar to peptide ^26^[M], its analogs exhibited a concentration-dependent cytotoxic effect against HTB-132 cells. All the analog peptides exhibited a significant cytotoxic effect against this cell line; however, the ^26^[D] and ^26^[K] peptides in which ^26^Met was replaced by charged amino acids (Asp and Lys, respectively) exhibited a lower cytotoxic effect against HTB-132 cells than the original dimeric peptide (Figure 1A). On the other hand, the ^26^[L] and ^26^[F] peptides exhibited a greater cytotoxic effect against cancer cells than the original peptide ^26^[M], indicating that when the Met was replaced with an amino acid containing a side chain of a hydrophobic-aliphatic or aromatic nature, the cytotoxic effect increased (Figure 1B).

Figure 1C shows the cytotoxic effect of the peptides at 100 μg/mL. The analog peptides ^26^[L] and ^26^[F] exhibited the highest cytotoxic effect against HTB-132 cells. The maximum cytotoxic effect was observed when the peptide concentration was 200 μg/mL. Peptides ^26^[L] and ^26^[F] decreased cell viability to 17% and 11%, respectively.

According to our results, peptides ^26^[L], ^26^[A], and ^26^[F] were chosen for evaluating their cytotoxic effect against the MCF-7 cell line (Figure 1D–F). As can be seen, these three peptides had a greater cytotoxic effect against MCF-7 cells than the peptide ^26^[M], which is similar to the cytotoxic effect of these peptides against HTB-132 cells. These peptides exhibited the same pattern of cytotoxic effect against both cell lines: the cytotoxic effect of peptides increased in the following order: ^26^[F] > ^26^[L] > ^26^[A] > ^26^[M]. Peptides ^26^[L] and ^26^[A] exhibited a similar cytotoxic effect, while the peptide ^26^[F] exhibited the highest cytotoxic effect against both cell lines. MCF-7 cells were more susceptible to analog peptides than HTB-132 cells, while the original peptide did not exhibit a cytotoxic effect against MCF-7 cells (Figure 1D). The IC_50_ values of peptide ^26^[F] for HTB-132 and MCF-7 cells were 13 µM and 6 µM, respectively, while for peptide ^26^[L] the IC_50_ values were 15 µM and 20 µM, respectively. These results allow us to consider the dimeric peptides ^26^[F] and ^26^[L] to be promising for the development of drugs for use against breast cancer, which is in agreement with previous reports that suggest that the cytotoxic effect of a molecule can be considered relevant when it has IC_50_ values below 25 µM [21].

The cytotoxic effect of the peptides ^26^[L] and ^26^[F] against MCF-7 cells were evaluated at 2, 24, and 48 h of treatment (Figure 2A). Peptides ^26^[L] and ^26^[F] exhibited a significant cytotoxic effect after 2 h of treatment when the peptide concentration was higher than 100 µg/mL. The cytotoxic effect of the peptides was concentration-dependent in all cases, and there were no significant differences in the cytotoxic effect among the evaluated treatment times. Furthermore, the cytotoxic effect of both peptides against MCF-7 cells was sustained for up to 48 h. Regardless of the treatment time, the maximum cytotoxic effect was observed when the peptide concentration was 200 µg/mL, which for peptide ^26^[L] was close to 85% (cell viability 15%), while for peptide ^26^[F] it was approximately 90% (cell viability 10%) (Figure 2A). When cells were treated with peptide concentrations below 100 µg/mL the cytotoxic effect reached its maximum value at 24 h, decreasing when cells were treated for 48 h. This behavior is possibly due to the fact that peptide concentration in the culture medium decreased allowing cell proliferation. On the other hand, for cells treated with a peptide concentration equal to or higher than 100 µg/mL the cytotoxic effect was maintained up to 48 h.

To determine if the cytotoxic effect of the peptide is selective for breast cancer cells, the human immortalized non-tumor epithelial cell line MCF-12 was treated with the peptides ^26^[F] and ^26^[L]. This cell line has been used as a control of normal cells in previous studies of breast cancer [22,23]. The peptides ^26^[L] and ^26^[F] exhibited a lower cytotoxic effect against MCF-12 cells than against breast cancer cell lines HTB-132 and MCF-7 (Figure 2B). The peptide ^26^[L] exhibited a selective cytotoxic effect in a peptide concentration range between 6 and 100 µg/mL, The IC_50_ values for peptide ^26^[L] in HTB-132 (10 µM/32 µg/mL) and MCF-7 (20 µM/26 µg/mL) was less than the IC_50_ value in the non-tumorigenic cell line MCF-12 (>100 µg/mL). The peptide ^26^[F] exhibited the greatest selectivity for MCF-7 cells when the peptide concentration was between 6 and 50 µg/mL. The IC_50_ values for peptide ^26^[F] in HTB-132 (13 µM/43 µg/mL) and MCF-7 (6 µM/19 µg/mL) was less than the IC_50_ value in the non-tumorigenic cell line MCF-12 (21 µM/ 70 µg/mL). When the peptide concentration was 6 µM, the cell viability for MCF-7 cells was 50%, while at the same peptide concentration, the cell viability for MCF-12 cells was near 100%. These results are consistent with previous studies, where the selectivity of LfcinB analogs against other non-tumorigenic cell lines has been demonstrated, which implies that these peptides are promising molecules for subsequent ex vivo or in vivo studies and the future development of drugs for use against breast cancer [15,16,17,24].

Cell morphology was monitored before and after treatment with ^26^[L] or ^26^[F] at different times, using an inverted microscope coupled to an AxioCam ICc1 camera (Figure 3). The untreated cells had normal morphologic characteristics as polygonal, flattened, and elongated cells with long and defined axons (data not shown) [25]. After 5 min of treatment with the peptide, most cells retained their morphology; however, some began to appear rounded and shrunken. After 90 min, the cells treated with the peptide (^26^[L] or ^26^[F]) took on a rounded shape, and shrinkage was observed, going from an average size of 100.27 μm to 39.38 µm. However, the cells maintained a defined cell morphology, and membrane integrity apparently was not compromised, suggesting that the cytotoxic effect of the peptides could involve apoptotic processes [26]. Also, this behavior was observed at 4 h of treatment. At 24 h of treatment, the appearance of small vacuoles was observed, presumably associated with late apoptosis events. Finally, it was observed that the effect of the peptide was sustained up to 48 h since at that time no recovery of cell morphology was observed. These results indicate that the cytotoxic effect of the peptides on MCF-7 cells evaluated in MTT assays could be associated with these observed morphologic changes.

With the aim of establishing if the cytotoxicity of peptide ^26^[F] is associated with loss of cytoplasmic membrane integrity, MCF-7 cells were treated with the peptide in the presence of propidium iodide (PI) and SYTO9 (Figure 4A). When the cells were treated with the peptide at 15 µM (IC_50_), it was determined that only 10% of the cell population had their cytoplasmic membrane affected, which is in accordance with the morphological changes that showed that the membrane continues to be preserved after treatment, indicating that the type of cell death may be being mediated by apoptotic pathways.

To determine the type of cell death associated with the cytotoxic effect of the ^26^[F] peptide on MCF-7 cells, they were incubated with the peptide (15 µM) for 24 h in the presence of annexin V and PI fluorophores (Figure 4B). As can be seen, 54% of the cell population was dyed with both fluorophores (cells in an apoptotic process) and 21% was dyed with only PI (cells in a necrotic process). These results indicate that the cytotoxic effect of the peptide against MCF-7 cells generates a higher cell population involved in late apoptotic events. These results are in agreement with those obtained previously here: changes in the cellular morphology such as shrinking, the appearance of dendritic bodies, and low affectation of the integrity of the cytoplasmic membrane, which could be related to death via apoptosis.

The mitochondrial membrane depolarization in MCF-7 cells treated with peptide ^26^[F] (15 µM) in the presence of cationic fluorophore JC-1 was evaluated (Figure 4C). The results showed that the peptide induces depolarization of the mitochondrial membrane in 19% of the population, a value even higher than that exerted by the positive control of apoptosis (ActD 15 µM). These results suggest that peptide ^26^[F] induces cell death in MCF-7 cells via intrinsic apoptosis, which is in agreement with the results previously obtained in the prior flow cytometry assays using PI/annexin V fluorophores. This result is similar to another one found in previous studies with a tetrameric molecule derived from LfcinB [17]. In addition, they are in accordance with results obtained by other authors that suggest that BLF, LfcinB, and synthetic peptides derived from LfcinB exhibited a selective, fast, and concentration-dependent cytotoxic effect against diverse human cancer cell lines, including breast cancer cell lines, apoptosis being the action mechanism proposed [11,12,27,28,29,30,31,32,33,34,35,36,37,38,39,40,41,42].

## 3. Discussion

In previous reports, we showed evidence that the polyvalence of LfcinB sequences significantly enhances the cytotoxic effect in oral and breast cancer cell lines: dimeric and tetrameric peptides containing the minimal motif exhibited a selective, rapid, and concentration-dependent cytotoxic effect against breast cancer cell lines. In addition, it was possible to establish the viability of the synthesis of these polyvalent peptides via SPPS, which allows modifications to be made in the sequence in order to identify new peptides with a greater cytotoxic effect against breast cancer cell lines [16]. Within this context, the change in the 26th position (^26^Met) of the dimeric peptide LfcinB (20-25)_2_ was evaluated.

It should be noted that the change of ^26^Met to ^26^Lys increased the positive net charge of the peptide to +14. However, the cytotoxic effect did not increase, suggesting that the positive charge of the peptide is not the only requirement for exerting the cytotoxic effect against the HTB-132 cells. On the other hand, the fact that the incorporation of hydrophobic residues instead of Met increased the cytotoxic effect against breast cancer cell line HTB-132 and MCF-7 is in agreement with previous reports that suggest that a hydrophobic amino acid such as Trp is a relevant residue in peptide activity because it is involved in membrane disruption and/or cell internalization [43,44]. Interestingly, the change from Met to Leu or Phe significantly increased the cytotoxic effect against this breast cancer line. Both these residues have a lower polarity than Met and are flanked by positively-charged side chains of ^25^Arg and ^27^Lys, which could increase peptide amphipathicity. This is in accordance with reports suggesting that the amphipathicity of LfcinB-derived sequences is relevant for antibacterial and anticancer activity [16,38]. Interestingly, our results indicate that position 26 of the dimeric ^20^RRWQWRMKKLG^30^ sequence is relevant for the cytotoxic effect.

Our results are exceptionally relevant because these dimeric peptides exhibited a selective cytotoxic effect against breast cancer cell lines better than BLF, which also exhibited inhibitory effects on the growth of four breast cancer cell lines, T-47D, MDA-MB-231, Hs578T, and MCF-7, but not for the normal breast cell line MCF-10-2A [34,39,41,45]. LfcinB showed considerable potential as an anti-cancer agent because the peptide is able to trigger apoptosis in a wide range of breast cell lines (MDA-MB-435, MDA-MB-231, T-47D, and MCF-7) without damaging normal human cells such as lymphocytes, erythrocytes, endothelial cells, fibroblasts, and breast epithelial cells [37,40,46]. Over the last decade, great efforts have been made to identify short LfcinB-derived peptides with increased antibacterial and/or anti-cancer activity [46].

Dimeric peptides ^26^[F] and ^26^[L] containing the minimal motif (RRWQWR) exhibited a significant cytotoxic effect against the tested breast cancer cell lines, which agrees with previous studies that showed that short synthetic peptides containing this motif also exhibited a cytotoxic effect against breast cancer cell lines [47]. The peptide RRWQWR exhibited a low cytotoxic effect against breast cancer cell lines MDA-MB-231 and MDA-MB-468 [16,37,38]. However, when this motif was modified or included in longer sequences, the cytotoxic effect against human cancer cell lines increased. In this context, we wish to emphasize the polyvalent peptides that contained the RWQWRWQWR, RRWQWRMKKL, and RRWQWR sequences, which showed significant cytotoxic activity against breast and oral cancer cell lines [15,16,17,24]. In addition, we have demonstrated the viability of the synthesis of dimeric peptides ^26^[F] and ^26^[L], which is easier, faster, and lower-cost than BLF or LfcinB, this being an advantage for drug development. On the other hand, both dimeric peptides contain the unnatural amino acid Ahx, which can confer greater stability to proteolytic degradation and increase the effective peptide concentration in the body.

LFB and LfcinB are candidates for alternative cancer treatment with the same advantages but without the side effects that characterize standard therapies [45]. A major limitation of the therapeutic efficacy for cancer is usually the systemic bio-distribution, which often leads to reduced bioavailability of the drug delivered to the cancer cells, and consequently to a reduction in the therapeutic index [45].

In this investigation we identified two dimeric peptides, (RRWQWR**F**KKLG)_2_-K-Ahx and (RRWQWR**L**KKLG)_2_-K-Ahx, that exhibited a selective cytotoxic effect against MCF-7 cells, which may be associated with apoptotic processes. Their synthetic viability, their dimeric structure, and the unnatural amino acid are advantages that allow us to consider them to be promising molecules for further studies in the development of therapeutic alternatives for breast cancer. Position 26th of the LfcinB (20-30) plays a critical role in the cytotoxic effect of the dimeric peptide LfcinB (20-30)_2_ against breast cancer cell lines. The insertion of hydrophobic amino acids in this position dramatically improves the anticancer activity against breast cancer cell lines HTB-132 and MCF-7, resulting in faster action times (less than 90 min) sustained up to 48 h. In addition, the peptides were selective compared to the non-tumorigenic line MCF-12. The most promising peptide was that in which the M of position 26 was replaced by F. In this peptide, an IC_50_ of 6–13 µM was found, as well as selectivity against the MCF-12A cell lines. It was found via flow cytometry that ^26^[F] does not compromise the integrity of the cytoplasmic membrane; it exerts its effect through apoptosis, and intrinsic apoptosis events are involved in the type of cell death.

## 4. Materials and Methods

### 4.1. Reagents and Materials

The HTB-132 MCF-7 and MCF-12 cell lines were obtained from ATCC^®^ (Manassas, VA, USA); LIVE/DEAD^TM^ BacLightTM Viability Kit, and annexin V, Alexa Fluor^TM^ 488 conjugate was purchased from Invitrogen (Eugene, Oregon); Mitochondrial Membrane Potential Detection was obtained from BD Biosciences (Torreyana Rd., San Diego, CA 92121, USA). The DMEM culture medium was obtained from Sigma-Aldrich (St. Louis, MO, USA). Bovine fetal serum was purchased in Gibco (Waltham, MA, USA). The Fmoc-amino acids, Rink amide resin, dicyclohexylcarbodiimide and 1-hydroxy-6-chlorobenzotriazole were purchased from AAPPTec (Louisville, KY, USA). Acetonitrile, trifluoroacetic acid, dichloromethane, diisopropylethylamine, N, N-dimethylformamide, ethanedithiol, isopropanol, methanol, triisopropylsilane were purchased from Merck (Darmstadt, Germany). SPE SupelcleanTM columns were purchased from Sigma-Aldrich (St. Louis, MO, USA) and Silicycle^®^ SiliaPrepTM C18 columns were kindly donated by EcoChem Especialidades Químicas (Waterloo, QC, Canada).

### 4.2. Solid Phase Synthesis SPPS

The peptides were synthesized using solid-phase peptide synthesis (SPPS-Fmoc/tBu) [48,49]. Briefly, a Rink amide resin (0.46 meq/g) was used as a solid support. On this support, an elongation of the peptide sequences was performed by successive steps of (i) deprotection of the alpha-amino group, removing the Fmoc group under basic conditions with 2.5% 4-methylpiperidine (% *v/v*), (ii) activation of the Fmoc-amino acid with DCC and 6-Cl-HOBt in DMF and DCM (3:1) and adding 2 drops of Triton-X, and (iii) coupling the amino acid to the growing solid support chain by performing a microwave-assisted reaction (time reaction 10 s (5×), 200 W). The peptides were subsequently cleaved from the resin by adding TFA/H_2_O/TIS/EDT (92.5/2.5/2.5/2.5 % *v*/*v*) and stirring for 8 h, and then the peptides were precipitated using diethyl ether at −20 °C, dried at room temperature, and characterized via RP-HPLC and MALDI-TOF MS (detailed information available in Appendix A).

### 4.3. RP-HPLC Characterization

For the analysis of the peptides (1 mg/mL) by RP-HPLC, solvent A: 0.05% TFA in water and solvent B: 0.05% TFA in ACN was used as the mobile phase. As a stationary phase, a Chromolith^®^ C-18 monolithic column (50 × 4.6 mm) was used. An elution gradient of 5% to 50% of solvent B was used for 8 min at a flow rate of 2.0 mL/min, injection volume 10 µL, and 210 nm for detection. An Agilent Series 1260 chromatograph was used. The chromatographic profile of purified dimers is presented in Appendix A.

### 4.4. Peptide Purification by Solid Phase Extraction (SPE)

For the purification of the peptides, 5 g RP-SPE columns (particle size: 40–60 µm) were used [50]. The peptide was dissolved in solvent A, and the sample was seeded and then eluted with solutions containing different percentages of solvent B. The fractions containing the pure peptide were collected and lyophilized. The final products were stored at −4 °C.

### 4.5. MALDI-TOF Mass Spectrometry Analysis

The molecular weight of the peptides was determined via MALDI-TOF mass spectrometry (microFlex, Bruker). One microliter of the purified peptide solution (0.5 mg/mL) was mixed with 18 µL of matrix (α-cyano-4-hydroxycinnamic acid, 5 mg/mL), and then 1 µL of the mixture was seeded on the plate sample holder. The laser power ranged between 2700 and 3000 V, and 200 laser shots were made. The MALDI TOF MS spectra of the purified dimers are presented in Appendix A.

### 4.6. Cell Culture

For all cell lines, the medium used was Dulbecco’s Modified Eagle’s Medium (DMEM)/Nutrient Mixture F-12 Ham. For the HTB-132, MCF-7, and MCF-12A lines, the medium was supplemented with 10% fetal bovine serum (SFB), and 1.5 g/L NaHCO_3_ and NaOH were added up to pH 7.4, amphotericin (200 μg/mL), and 1% penicillin and streptomycin. For the primary culture cells of bovine mammary epithelium and fibroblasts, in addition to the above, hydrocortisone (250 μg/mL) was added. All media were filtered through a 0.22 μm membrane.

### 4.7. Viability Test by MTT

Briefly, the cells were seeded with a complete medium in 96-well plates at a rate of 10,000 cells and 100 µL per well, and adhesion to the plates was allowed for 24 h. Subsequently, the complete medium was removed and an incomplete medium was added for synchronization for another 24 h. The cells were then incubated at 37 °C for 2, 24, or 48 h with 100 μL of peptide at the concentrations to be evaluated (200, 100, 50, 25, 12.5, 6.25 and 3.1 μg/mL). Next, the peptide was removed from the box, and 100 µL of incomplete medium with 10% 3-4,5-dimethylthiazol-2-yl-2,5-diphenyltetrazole (MTT) bromide was added and incubated for 4 h. The medium was replaced with 100 μL of isopropanol (IPA), and after 30 min of incubation at 37 °C, the absorbance was measured at 575 nm. As a negative control, an incomplete culture medium with 10% MTT was used, and as a positive control, cells without MTT treatment were used [51].

### 4.8. Evaluation of the Integrity of the Cytoplasmic Membrane Using SYTO9/PI

The cells with complete medium were seeded and synchronized in boxes of 24 wells in a concentration of 4 × 10^4^ cells/well in a volume of 400 μL/well, and adhesion to the plate was allowed for 24 h [52]. Then the complete medium was replaced by incomplete medium for synchronization of the cells for an additional 24 h. Subsequently, the culture medium was replaced with 400 μL of a solution containing the peptide to be evaluated at a concentration equivalent to its IC_50_ and incubated for 2 or 24 h. The cells were harvested by adding 200 μL of trypsin and incubating for 10 min. The trypsin was quenched with 200 µL of complete medium, and the cells were centrifuged at 2500 rpm for 10 min. The supernatant was discarded, and the pellet was washed with 100 µL PBS and centrifuged under the same conditions previously described. The supernatant was discarded, and the cells were stained with 30 μL of a solution from the LIVE/DEAD^®^ FungaLigthTM commercial kit (Invitrogen) containing the mixture of fluorochromes (0.5 μL of SYTO9 and/or 0.5 μL of propidium iodide (PI) with 99 μL of PBS). Subsequently, the cells were incubated at room temperature and in the dark for 20 min and centrifuged, and the supernatant was discarded. The pellet was resuspended in 100 μL of PBS and analyzed via flow cytometry in a BD Accuri C6 device. The events were recorded using the green channel (FL1) on the abscissa axis and the red (FL3) on the ordinate axis. Negative control: untreated cells marked with both fluorophores; positive control: cells treated with actinomycin 10 μg/mL for 24 h [52]. To define the working population, a control of untreated and unstained cells was used. Two controls were used to compensate: (i) untreated cells stained only with PI, and (ii) untreated cells stained only with Syto9.

### 4.9. Determination of the Type of cell Death (Apoptosis/Necrosis)

The cells were seeded and synchronized in boxes of 24 wells at a concentration of 4 × 10^4^ cells/well in 400 μL/well, adhesion and synchronization were allowed in the same way as in Section 4.8, and the culture medium was replaced with medium containing the peptide to be evaluated and incubated for 2 or 24 h. Subsequently, the cells were harvested with trypsin, centrifuged at 2500 rpm for 10 min, washed with PBS, and resuspended in 10 μL of staining buffer with the fluorochromes (10 mM Hepes pH 7.4; 10 mM NaCl and 2.5 mM CaCl_2_ containing 1 μL of PI fluorochrome and 1 μL of Annexin V). Cells with the fluorochromes were incubated at 37 °C in the dark for 15 min and resuspended in 80 μL of staining buffer without fluorochromes for analysis via flow cytometry. The positive control of necrosis was: cells treated with EDTA 15 mM for 60 min, and apoptotic: cells treated with Actinomycin at 10 μM for 24 h. Negative control: cells without treatment; compensation controls: (i) cells stained only with Annexin, and (ii) only with PI; population control: unstained and untreated cells.

### 4.10. Determination of Mitochondrial Membrane Depolarization

For this test, the MitoProbeTM JC-1 Assay kit (M34152 from Termofisher, Walthan, MA, USA) was used according to the supplier’s recommendations. The cells seeded and synchronized in a box of 24 wells (4 × 10^4^ cells/well) were incubated with 400 μL of peptide to be evaluated at 2 and 24 h. The cells were then trypsinized and collected by centrifugation at 400 g × 5 min. The resulting pellet was stained by adding 100 μL of JC1 “working solution” (1:100 JCI solution reconstituted in DMSO: buffer assay 1×) and incubated at 37 °C for 20 min. Subsequently, the cells were washed twice with 1× buffer assay and finally resuspended in 100 μL of 1× buffer assay for reading on the cytometer. Negative control: cells without treatment; positive control: cells treated with 10 μM actinomycin for 24 h.

## Figures and Tables

**Figure 1 ijms-21-04550-f001:**
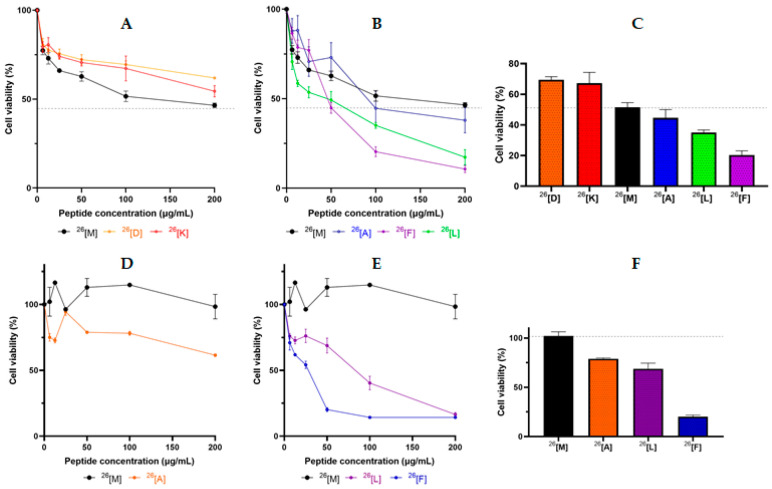
Cytotoxic effect of ^26^[M] (LfcinB (20-30)_2_) and its analogs against HTB-132 (**A**,**B**), and MCF-7 (**D**,**E**) cell lines. Breast cancer cells were treated with the peptide for 24 h at 37 °C. Comparative cytotoxic effect of peptides at 100 µg/mL against HTB-132 (**C**) and 50 µg/mL MCF-7 (**F**) cells. The data in **A**–**C** represent the mean ± SE (*n* = 3). The data in **D**–**F** represent the mean ± SE (three independent experiments with *n* = 4 each). Statistically significant differences were found between the cytotoxic effect exhibited by the modified dimeric peptides and that of the unmodified peptide ^26^[M]. (ANOVA, Sidak’s multiple comparisons test was used, *p* ≤ 0.05).

**Figure 2 ijms-21-04550-f002:**
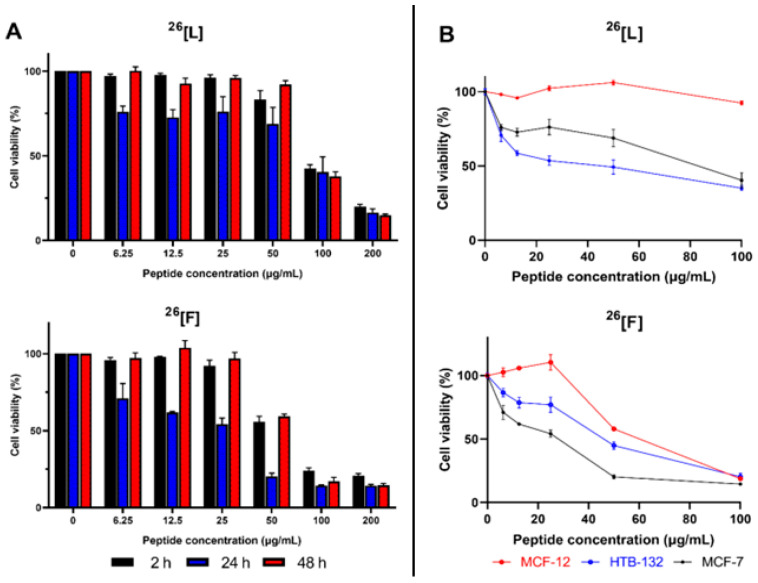
Cytotoxic effect of peptides ^26^[L] and ^26^[F] against MCF-7 cell lines. The MCF-7 cells were treated with the peptides for 2, 24, or 48 h at 37 °C (Panel **A**). Comparative cytotoxic effect of peptides ^26^[L] and ^26^[F] against MCF-7, HTB-132 and MCF-12 Cell lines; cells were treated with the peptides for 24, at 37 °C (Panel **B**). The data represent the mean ± SE (Three independent experiments with *n* = 4 each) Statistically significant differences were found in the cytotoxic effect against the breast cancer lines and the non-tumorigenic line MCF-12 in a range of 0 to 100 µg/mL for ^26^[L] and in a range of 0 to 50 µg/mL for ^26^[F]. (ANOVA, Sidak’s multiple comparisons test was used, *p* ≤ 0.05).

**Figure 3 ijms-21-04550-f003:**
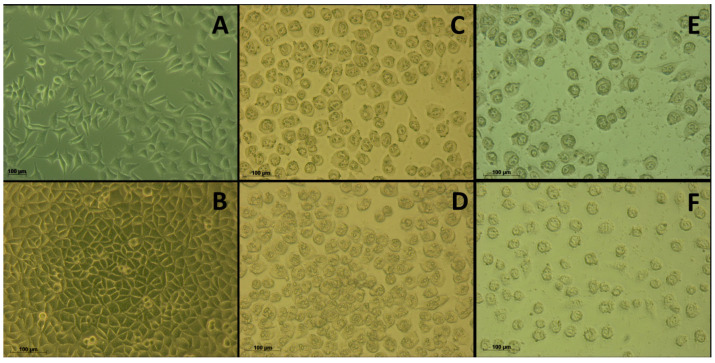
Micrography of MCF-7 cells treated with peptide ^26^[F] at 100 µg/mL. The cells were incubated in the absence (**A**) or the presence of the peptide at (**B**) 5 min, (**C**) 90 min, (**D**) 240 min, (**E**) 24 h, and (**F**) 48 h at 37 °C. The cell size was determined as an average of 100 measurements, using an AxioCam ICc1 camera.

**Figure 4 ijms-21-04550-f004:**
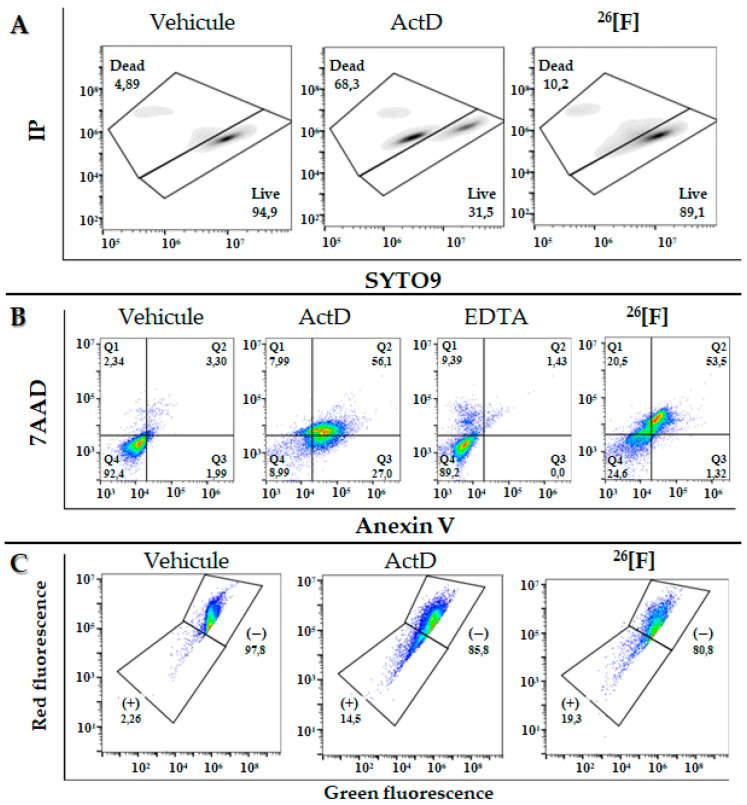
Flow cytometry assays. (**A**) MCF-cells treated with the peptide ^26^[F] (15 µM) in the presence of PI and SYTO9 fluorophores. Negative control: cells without treatment, vehicle (DMEM); positive control: cells treated with Actinomycin-D for 24 h. (**B**) MCF-7 cells treated with the peptide ^26^[F] in the presence of Annexin V and PI fluorophores for 24 h. Negative control: untreated cells, vehicle (DMEM); necrosis control: cells treated with 5 mM EDTA for 60 min; apoptosis control: cells treated with 15 µM Actinomycin-D for 24 h. (**C**) MCF-7 cells treated with the peptide ^26^[F] in the presence of JC-1 fluorophore. Negative control: untreated cells, vehicle (DMEM); positive control: cells treated with 15 µM Actinomycin-D for 24 h.

**Table 1 ijms-21-04550-t001:** Dimeric peptides derived from Bovine Lactoferricin (20-30). General structure, characterization and cytotoxic effect against breast cancer cell lines. The peptide chain amino acid at position 26 is shown in red.

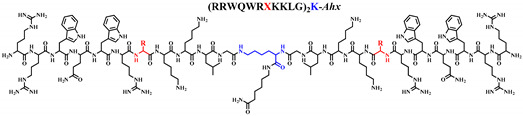						
Code	Sequence	Characterization			Cytotoxic Effect	
		RP-HPLC		*m*/*z* [M+H]^+^	IC_50_ (µM)/(µg/mL)	
		t_R_ (min)	Purity ^a^ (%)		HTB-132	MCF-7
^26^[M]	(RRWQWRMKKLG)_2_-K-*Ahx*	5.3	94	3312.3	30/96	>60/>200
^26^[K]	(RRWQWRKKKLG)_2_-K-*Ahx*	4.8	95	3302.7	>60/>200	N/D
^26^[D]	(RRWQWRDKKLG)_2_-K-*Ahx*	4.8	97	3276.7	>60/>200	N/D
^26^[A]	(RRWQWRAKKLG)_2_-K-*Ahx*	4.8	95	3187.9	26/96	>60/>200
^26^[F]	(RRWQWRFKKLG)_2_-K-*Ahx*	5.6	91	3342.5	13/43	6/19
^26^[L]	(RRWQWRLKKLG)_2_-K-*Ahx*	5.5	92	3272.4	10/32	20/66

^a^ Purity was calculated from the purified dimer chromatographic profile, Appendix A.

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
