# Peer review of "Peptides Derived from (RRWQWRMKKLG)2-K-Ahx Induce Selective Cellular Death in Breast Cancer Cell Lines through Apoptotic Pathway"

_ijms, 2020, doi:10.3390/ijms21124550_

Round 1

Reviewer 1 Report

  1. The supplementary could not be found. In addition, the labeling of Figure S1 and Figure 2S is not consistent.
  2. Have the authors tried to measure the cytotoxic effects of the proposed peptides against normal cells? 
  3. What are the possible mechanisms for this anti-breast cancer using the Bovine Lactoferricin-derived dimeric peptide?
  4. Further animal tests need to be performed in order to confirm the anti-cancer activity because there still is a huge gap between cell line test and animal test.

Reviewer 2 Report

The paper from Insuasty-Cepeda and coworkers characterize a dimeric peptide consisting of two 11-residue peptides LfcinB20–30 attached by means of Lys, as well as its five analogues. The cytotoxic effects of these peptides on breast cancer cell lines HTB-132 and MCF-7 is described. In the analogues, Met residue found in the parent LfcinB20–30 peptide was replaced with various amino acids – Lys, Asp, Ala, Phe and Leu. Such modification resulted in novel dimeric peptides – some of them exerted higher cytotoxicity on cancer cells and lowered toxicity towards non-tumorigenic cell line MCF-12.

The presented results are interesting. The authors shows good knowledge of the methodologies.

However, there are some points that should be addressed: 

  1. Unfortunately, I couldn’t find any information about the reason of Met replacement. What was the rationale to choose Met instead of other neighbouring resides? Does Met play a key role in antitumor activity in LfcinB20–30 or, which is more likely, it was due to its low stability and high sensitivity to be oxidized? The authors must provide such explanation.
  2. In the abstract, the following statement is present “The process of the synthesis of the LfcinB (20-30)2 analogue peptides was easier than for the original peptide.” I wasn’t able to find any evidences that might support that statement. Moreover, in the results section (line 80) we can read something that can undermine the above-mentioned declaration “…the substitution of amino acids does not affect the synthesis of the peptides”. So, did the substitution of Met residues have an impact on the chemical synthesis or not?
  3. After the statement (line 53) “Additionally, the dimeric peptide LfcinB (20-30)2: (20RRWQWRMKKLG30)2-K-Ahx has exhibited a significant selective cytotoxic effect against MDA-MB 468 and MDA-MB 231 breast cancer cells” the appropriate reference must be provided (I guess [16]).
  4. Line 86 “As is seen in Figure 1A, the peptide 26[M] exhibited a concentration-dependent cytotoxic effect against the HTB-132 cell line…” I would rather say that concentration-dependent cytotoxicity is reported in the 1 – 100 μg/mL range of concentration. In the range 100 – 200 μg/mL the observed decrease in cell viability is rather negligible.

5 Table 1 The primary structure of peptide (as well as in the Supplementary information Fig. S1) – there is Orn residue instead of Lys in position 8. It must be corrected.

Does the term Purity refers either to the crude peptide or to the one obtained after RP-SPE purification? Moreover, it should be stated how the Purity was calculated.

  1. Line 100. Regarding the statement “On the other hand, the 26[A], 26[L], and 26[F] peptides exhibited a greater cytotoxic effect against cancer cells than the original peptide 26[M]… “ Based on the Fig. 1B, the cytotoxicity of 26[A] on HTB-132 cells is only insignificantly higher (and only at the concentrations higher than 100 μg/mL) as compared to 26[M] and therefore, in my opinion, it shouldn’t be mention together with 26[F] and 26[L], which are actually more potent than 26[M]. I would suggest to describe 26[A] separately from 26[F] and 26[L].
  2. Fig. 1F I wonder, why the bar associated with 26[L] reaches almost 75% of cell viability. Comparing with the graph on Fig. 1E, this value seems to be lower (about 50%). If it is true, the statement in line 121 must be revised “Peptides 26[L] and 26[A] exhibited a similar cytotoxic effect, while the peptide 26[F] exhibited the highest cytotoxic effect against both cell lines”.
  3. Regarding Fig. 2A, the authors should make an attempt to explain the reported lower cell viability after treatment with peptide at the concentrations lower than 100 μg/mL after 24 h of incubation in comparison with both, shorter (2 h) and longer (48 h) incubation periods.
  4. Line 146 There is the statement “This peptide exhibited a lower cytotoxic effect against MCF-12 cells than against breast cancer cell lines HTB-132 and MCF-7 (Figure 2B).” For better clarity, the symbol of this peptide, 26[L] , should be included in this statement. Moreover, I suggest to express selectivity by calculating the selectivity indexes for both peptides (IC50 calc. for normal / IC50 calc. for cancer)
  5. Line 171 “.. the cytotoxic effect of the peptides on MCF-7 cells evaluated in MTT assays could be associated with these observed morphologic changes” In my opinion, it would be interesting to show how 26[M] affects (or doesn’t affect) cell morphology.
  6. Line 297 The conditions applied during microwave-assisted reaction (time, temperature) should be provided.
  7. Line 297 The peptides were rather cleaved from the resin, not separated.
  8. Line 358 The multiplication sign must be used in place of the letter x (4x104 cells/well). This applies to all manuscript.
  9. Line 359 “… was allowed in the same way as in 9.15” What does 9.15 mean?

Supplementary information

Figure 1 is pretty illegible. I would suggest to use only small letters (not STEP1, 2 etc) to point at the following steps during peptide synthesis. Their meanings should be included in the Figure description.

Line 14 It should be methionine instead of Metionine

Line 23 Correct the word reacción

Figure S2. Please, provide the applied HPLC conditions in the Figure description.

The MS spectra must be provided.

Reviewer 3 Report

This is a fine manuscript to study and optimize the LfcinB peptide cytotoxic effect on breast cancer lines. The authors synthesized a series of LfcinB peptide analogues and tested their cytotoxic effects. The authors found several mutations derivatives had improved potency compared to the parent one. The authors also studied the mode of action by microscope and flow cytometry to investigate their cellular morphology and cell apoptosis respectively. The paper has some values, but a minor revision is required. 

  1. There are twenty AA residues in the dimer LfcinB(20-30), and the authors simply changed 26Met to other AA. Is there a specific reason or rationale to mutate this residue? 
  2. Do we know the target for this peptide? 
  3. The authors found that mutations to L/F decreased cell viability. What could be the possible underlying mechanism? 
  4. How soluble are these peptides? 
  5. Fig. 1, C and F letters are out of places. Overall this figure should be rearranged and resized nicely. There are extra while spaces between graphs. 
